# Liver Regeneration and Cell Transplantation for End-Stage Liver Disease

**DOI:** 10.3390/biom11121907

**Published:** 2021-12-20

**Authors:** Yan Li, Lungen Lu, Xiaobo Cai

**Affiliations:** Department of Gastroenterology, Shanghai General Hospital, Shanghai Jiaotong University School of Medicine, Shanghai 200025, China; liyan19841231@sina.com (Y.L.); lungenlu1965@163.com (L.L.)

**Keywords:** end-stage liver disease, liver regeneration, massive hepatic necrosis, liver progenitor cells, cell transplantation

## Abstract

Liver transplantation is the only curative option for end-stage liver disease; however, the limitations of liver transplantation require further research into other alternatives. Considering that liver regeneration is prevalent in liver injury settings, regenerative medicine is suggested as a promising therapeutic strategy for end-stage liver disease. Upon the source of regenerating hepatocytes, liver regeneration could be divided into two categories: hepatocyte-driven liver regeneration (typical regeneration) and liver progenitor cell-driven liver regeneration (alternative regeneration). Due to the massive loss of hepatocytes, the alternative regeneration plays a vital role in end-stage liver disease. Advances in knowledge of liver regeneration and tissue engineering have accelerated the progress of regenerative medicine strategies for end-stage liver disease. In this article, we generally reviewed the recent findings and current knowledge of liver regeneration, mainly regarding aspects of the histological basis of regeneration, histogenesis and mechanisms of hepatocytes’ regeneration. In addition, this review provides an update on the regenerative medicine strategies for end-stage liver disease. We conclude that regenerative medicine is a promising therapeutic strategy for end-stage liver disease. However, further studies are still required.

## 1. Introduction

During recent decades, end-stage liver disease (ESLD) has been increasing in incidence. ESLD is the final stage of various liver diseases, which is associated with a high degree of mortality and presents a significant worldwide economic burden [1]. Currently the only curative therapy for ESLD is orthotopic liver transplantation (OLT); however, OLT generally brings related risks and requires lifelong immunosuppressive therapy. Furthermore, the shortage of donors and high cost might well limit the use of OLT [2]. Hence, alternative therapeutic strategies have been explored to mitigate the clinical challenge.

The liver is a highly regenerative organ. After injury, it is able to restore its mass and physiological functions [3]. Liver regeneration is prevalent in liver injury settings but generally compromised in severe liver diseases, hence it is assumed that promoting liver regeneration should be beneficial to patients with ESLD. A number of studies have concentrated on the mechanisms of liver regeneration. A better understanding of liver regeneration has led to improvements in therapeutic strategies for ESLD. In recent years, regenerative medicine, especially cell transplantation, has shown promises as alternatives to OLT in patients with ESLD. In this review, we mainly summarize the recent findings and current understanding of liver regeneration, regarding the histological basis, types and mechanisms of liver regeneration, as well as the current status and therapeutic potential of regenerative medicine for ESLD.

## 2. Histological Basis of Regeneration

Liver regeneration is a complicated process that involves the cooperation of various cells. The liver consists of parenchymal and nonparenchymal cells. Parenchymal cells are hepatocytes, while nonparenchymal cells mainly include hepatic stellate cells (HSCs), Kupffer cells (KCs), and liver sinusoidal endothelial cells (LSECs). Hepatocytes constitute 70% of total liver cells. As the major cell type in the liver, hepatocytes play a vital role in detoxification, metabolism, and coagulation [4]. KCs are the resident macrophages of the liver, representing about 20% of the nonparenchymal cells. They serve as the immune sentinels of the liver, regulating immune defense and phagocytosis through delicate cell–cell interaction and secreted cytokines. Aside from KCs, the liver is also enriched in numerous innate and adaptive immune cells, including natural killer cells (NK cells), natural killer T (NKT cells), neutrophils, γδT cells, dendritic cells (DCs), innate lymphoid cells (ILC), conventional αβT cells and B cells, which help to maintain the homeostasis of immune defense [5]. HSCs play a role in storing vitamin A and lipids; once injury occurs, HSCs might differentiate into myofibroblasts [4]. LSECs represent approximately 15 to 20% of the liver cells. They constitute liver sinusoid, with a fenestrated and discontinuous basement membrane. LSECs have unique functions, including fluid filtration, blood vessel tone modulation, blood clotting, inflammatory cell recruitment, and metabolite/hormone trafficking [6]. Besides, biliary epithelial cells (BECs) form the biliary network for bile transportation. BECs play an essential role in secretory and reabsorptive processes of bile, which subsequently influence the modification of bile composition and flow. All of the cells above can affect liver regeneration in different ways.

Liver progenitor cells (LPCs) are thought to play an important role in liver regeneration. LPCs arise near the canals of Hering, which connect the hepatocyte canalicular system and the biliary tree. LPCs are thought to be bipotential, and might differentiate into hepatocytes or BECs. It is suggested that LPCs generally express both hepatocyte markers (KRT8, KRT18, and et al) and BEC markers (KRT7, KRT19, EpCAM, SOX9, and et al). Additionally, LPCs might also express hepatoblast markers, hematopoietic markers, or even neuronal markers upon different injury settings. The diversity of these markers in LPCs suggests their progenitor features and heterogeneous nature [3]. However, the presence of LPCs in the human liver is still controversial, since identification methods for LPCs are not well-established.

## 3. The Typical Liver Regeneration

It has been demonstrated that the origin of hepatocytes during liver regeneration vary in different liver-damage models [7]. Hepatocytes are the main cells driving typical liver regeneration, whereas the alternative liver regeneration is driven by LPCs. 

The typical liver regeneration is achieved via the hypertrophy and/or hyperplasia of pre-existing hepatocytes, which is specific to healthy livers after massive liver mass loss. Partial hepatectomy models, first established by Higgins and Anderson in 1931, contribute to better elucidating the mechanisms of typical liver regeneration. The process of typical liver regeneration consists of three stages: priming phase, proliferation phase and termination phase. 

The priming phase is the initial phase of regeneration, during which the hepatocytes simultaneously enter the G1 phase of the cell cycle, driven by various cytokines. After two-thirds partial hepatectomy, the complement system is activated to initiate the regeneration process, triggering a number of cytokines to promote regeneration. Tumor necrosis factor-α (TNF-α) and interlukin-6 (IL-6) are the most important cytokines during the priming phase. TNF-α activates the NF-κβ signaling pathway and stimulates c-Jun N-terminal kinase (JNK), which phosphorylates the c-Jun transcription factor in the nucleus to induce cyclin-dependent kinase 1 transcription. IL-6 activates the Janus kinase (JAK)/signal transducer and activator of transcription (STAT), mitogen-activated protein kinase (MAPK) and PI3K/AKT signaling pathways. The quiescent hepatocytes enter the cell cycle (G0 to G1 phase) in response to the proregenerative signals [8]. 

During the proliferation phase, both complete and auxiliary mitogens converge upon hepatocytes/BECs to drive the G1/M phase transition. Complete mitogens are signals that are mitogenic for hepatocytes in chemically defined serum-free media in primary culture, and when administered to intact, unoperated animals cause liver enlargement. Complete mitogens mainly include hepatocyte growth factor (HGF) and epidermal growth factor receptor (EGFR)-associated ligands including epidermal growth factor (EGF), transforming growth factor-α (TGF-α), amphiregulin, as well as heparin-binding epidermal growth factor-like growth factor (HB-EGF). HGF interacts with the methionine receptor to activate PI3K/AKT and extracellular-signal-regulated kinase 1/2 signaling pathways. EGFR-associated ligands and EGFR interaction activates MAPK, PI3K/AKT and STAT signaling pathways. Auxiliary mitogens are not mitogenic in primary cultures of hepatocytes in chemically defined media, and their administration in intact animals is not associated with liver enlargement. Auxiliary mitogens mainly include norepinephrine, bile acids, leptin, serotonin, insulin, and others. Auxiliary mitogens control the timing of essential transcription factors associated with hepatocyte proliferation and synergize with complete mitogens. Deprivation of auxiliary mitogens would delay but not abolish liver regeneration. Proliferating hepatocytes produce a series of growth signals, including PDGF-A, VEGF, GMCSF, and others. In response to these growth signals, HSCs, KCs and LSECs start proliferating and secrete growth signals directed back to hepatocytes [9].

During the termination phase, the proliferating hepatocytes return to the differentiated, quiescent state upon completion of regeneration. When the needed liver size is achieved, the proliferation phase is stopped due to inhibitory molecules. Another function of the inhibitory molecules is ensuring the regeneration proceeds in the correct direction by preventing proliferation in wrong directions. Transforming growth factor-β (TGF-β) is the most significant inhibitory molecule in the process of liver regeneration. TGF-β modulates hepatocyte proliferation via inhibiting DNA synthesis induced by HGF, EGF and HB-EGF. Furthermore, TGF-β might inhibit secretion of HGF and induce apoptosis. Interlukin-1 (IL-1) significantly inhibits DNA synthesis induced by insulin and EGF in cultured rat hepatocytes. Interferon-γ (IFN-γ) can downregulate hepatocyte proliferation by activating STAT1 and the downstream genes. The suppressor of cytokine signaling (SOCS) family is a family of proteins playing important roles in the terminating phase. SOCS3 could inhibit the IL6/JAK/STAT3 pathways and proliferation evoked by HGF/EGF. SOCS1 has similar functions but does not inhibit EGF-mediated proliferation. Other terminating signals include bone morphogenetic proteins (BMPs), integrin-linked kinase, myeloid differentiation factor 88 (MyD88), and others [10]. IL-6 is a dual regulator of liver regeneration, depending on the time and dose. Huck et al. investigated the antiproliferative effect of hepatocyte nuclear factor 4α (HNF4α) in regulation of liver regeneration in mice hepatectomy models. HNF4α rapidly declines after hepatectomy and does not recover to the baseline level until day three, which enables the hepatocytes entry into the cell cycle during the priming phase of liver regeneration. The re-establishment of HNF4α activity is crucial for hepatocytes differentiation and termination of liver regeneration [11].

The immune cells are indispensable during the process of the typical liver regeneration. KCs produce TNF-α and IL-6, initiating the regeneration process. KCs depletion would greatly compromise liver regeneration [12]. DCs upregulate interlukin-10 (IL-10) expression level and downregulate IFN-γ expression level, which facilitates liver regeneration. It is demonstrated that T-cell deficiency would greatly compromise liver regeneration since the conventional αβT cells can secrete lymphotoxin and stimulate liver regeneration. Liver eosinophil-derived interlukin-4 (IL-4), γδT cell-derived interlukin-17 (IL-17), and ILC-derived interlukin-22 (IL-22) also promote the regeneration process. NK and NKT cells play inhibitory roles in liver regeneration via the IFN-γ pathways [5]. Furthermore, KCs’ repopulation after hepatectomy in mice is mainly driven by local KCs’ proliferation, which is dependent on IL-6 and (SIRT1) activation in KCs [13].

Apart from immune cells, other nonparenchymal cells are also important during the process of typical regeneration. During the priming phase, BECs secrete osteopontin and monocyte chemoattractant protein-1 (MCP-1), while LSECs secrete intercellular adhesion molecule 1 (ICAM-1). In addition, HSCs might secrete MCP-1. These chemotaxis mediators are crucial for macrophage recruitment to the liver [5,8].

## 4. The Alternative Liver Regeneration

Massive hepatic necrosis (MHN) is a key event underlying ESLD, which denotes an extensive multilobular/panacinar hepatocyte necrosis [14]. During the process of MHN, almost all parenchymal cells die; hence, the typical liver regeneration is greatly hampered. Instead, MHN rapidly induces the alternative liver regeneration mediated by LPCs [15]. The alternative liver regeneration is achieved by the proliferation and subsequent differentiation of LPCs in response to inflammatory cytokines (Figure 1), which is specific to MHN. LPCs can differentiate into BECs or hepatocytes to restore the liver mass and functions upon stimulus [8]. The alternative regeneration persists for a long time and helps restore liver parenchyma and functions. There is a positive correlation between parenchymal loss and LPCs’ proliferation. Katoonizadeh et al. examined liver specimens from 74 patients with acute or subacute severe liver impairment and revealed that 50% death of hepatocytes is a threshold for extensive activation of LPCs [16]. In conclusion, LPCs take over key hepatocyte functions in cases of massive liver injury, which ultimately determines survival [17]. 

As determined in various liver diseases in humans and animal models, there are hepatocytes with BECs markers that emerge ectopically in the parenchymal region of the liver [18,19,20,21]. These intermediate cells are thought to be derived from LPCs from biliary compartment. That is to say, a certain population of BECs dedifferentiates into LPCs upon injuries, which proliferate and eventually differentiate into hepatocytes. In certain liver-injury animal models, hepatocytes [22] and even HSCs [23] might be the source of LPCs.

The molecular mechanisms regulating LPCs differentiation play pivotal roles in liver regeneration, which has not been well-acknowledged until now. LPCs activation is the first step in the alternative liver regeneration. The homeostasis of LPCs relies on the balance of complex signals in their microenvironment. In a quiescent state, the microenvironment keeps LPCs as the progenitor phenotype and inhibits cell differentiation. Liver injuries trigger specific alterations of the microenvironment, promoting the differentiation of LPCs toward a hepatocyte phenotype or BEC phenotype regulated by a variety of signaling pathways [24]. VEGF is considered an important cytokine during the activation of LPC-mediated liver regeneration in patients with nonalcoholic steatohepatitis (NASH) [25]. The balance between Wnt and Notch signaling is considered a key mechanism to steer LPC differentiation in either direction. Prominent Wnt signaling drives LPC differentiation into hepatocytes, whereas Notch signaling drives LPC differentiation toward BECs [14]. Mehwish et al. demonstrated that in zebrafish the Stat3/Socs3a pathway is necessary for the proper timing of LPC-to-hepatocyte differentiation and establishing the proper number of BECs during the alternative liver regeneration [26]. Jacquelyn et al. reported that during choline-deficient, ethionine-supplemented (CDE) diet-induced liver injury, β-catenin deletion provokes BECs to differentiate into hepatocytes, which permanently incorporate into the liver parenchyma to mediate liver regeneration [27]. Sungjin et al. suggested that histone deacetylase 1 gene (HDAC1) regulates differentiation of LPCs into hepatocytes via Sox9b, and differentiation of LPCs into BECs via Cdk8, Fbxw7, and Notch3 in a liver-injury model of zebrafish [28]. Further study revealed that the EGFR–ERK–Sox9 axis suppresses LPC-to-hepatocyte differentiation in zebrafish, indicating EGFR inhibitors as a proregenerative therapeutic drug for patients with ESLD [29]. Yuki et al. reported that trefoil factor family 1 (TFF1) promotes LPCs’ differentiation into a biliary lineage, and inhibits LPCs’ differentiation into a hepatic lineage in mice [30]. Francesco et al. presented that the perturbation of redox balance by induction of a pro-oxidative environment may activate BECs-derived LPCs and nuclear factor (erythroid-derived 2)-like 2 (NRF2) is the main redox-dependent transcription factor driving LPC fate. NRF2 is constitutively activated in LPCs to maintain their stemness features, whereas it is inhibited in case of LPC activation. NRF2 inhibition favors engraftment and repopulation of damaged cells by transplanted elements, which increases the transplantation efficiency of LPCs [24]. A recent study in vitro demonstrated that autophagy regulates hepatic differentiation of LPCs through the Wnt signaling pathway [31]. Another study in vitro revealed that PPARγ acts as a negative regulator of liver regeneration by inhibiting the proliferation of LPCs, which induces cell cycle G0/1 phase arrest through the Hippo/YAP pathway [32]. Furthermore, farnesoid X receptor (FXR) has been identified as inhibitors of the alternative liver regeneration by enhancing phosphatase and tensin homolog (PTEN) activity. It has been confirmed in zebrafish that FXR activation blocks LPC-to-hepatocyte differentiation, but not BEC-to-LPC dedifferentiation [33]. Importantly, it is supposed that these pathways might be manipulated to induce LPC differentiation for treatment of patients with ESLD.

## 5. Regenerative Medicine and Cell Transplantation for ESLD

Advances in knowledge of physiology of liver regeneration and tissue engineering have accelerated the progress of regenerative medicine therapies for ESLD [34,35]. The regenerative medicine strategies may alleviate liver damage by replacing injured hepatocytes, stimulating proliferation of the pre-existing hepatocytes, and creating a growth-permissive microenvironment for the survival and integration of transplanted cells into the host. Cell transplantation is one of the most promising regenerative medicine strategies for treatment of ESLD. A wide variety of cell types can be transplanted to achieve liver regeneration enhancement [4].

### 5.1. Hepatocytes Transplantation

As mentioned above, the typical liver regeneration is achieved via the proliferation of pre-existing hepatocytes; however, it is generally compromised in ESLD due to MHN. Hence, supplement of exogenous mature hepatocytes might be a potential therapeutic target for treatment of ESLD. 

The first hepatocytes transplantation (HT) dates back to 1976, where it was used to treat hyperbilirubinemic rats [36]. Additionally, the first attempt of HT in humans was developed in 1992 for the treatment of cirrhotic patients [37]. During recent decades there has been a considerable development of HT, and standardized techniques have been established for isolation, culture, and cryopreservation of human hepatocytes. To date, allogeneic HT has been used in a number of liver diseases, with demonstration of short-term efficacy and safety. 

HT is a potential alternative to OLT. Compared to OLT, HT is less invasive and less expensive. Cell grafts are generally isolated from human livers that are unsuitable for transplantation, raising the possibility of using one donor organ for more recipients. Hepatocytes can be transplanted through intraportal, intrasplenic or intraperitoneal routes, which do not require complex surgery. If required, HT can be repeatedly performed. Cryopreserved cells isolated from donor livers are available immediately when needed. Additionally, HT may be beneficial to promote host parenchymal regeneration. Furthermore, the native liver remains in place, serving as a backup in case of cell graft failure [38]. 

However, clinical efficacy of HT is still controversial over the long term. The drawbacks of HT mainly include shortage of high-quality engraftment and histologic incompatibility. Cell numbers required to achieve physiological benefit has been estimated at around 5–15% of the oretical liver mass [39]. Hence, it is of vital importance to obtain adequate hepatocytes for transplantation. Until now, the yield and viability of hepatocytes during isolation, culture, cryopreservation and transplantation has not been satisfactory, and further improvement is required. Post-HT immunosuppressive therapy is necessary due to transplant rejection [40]. Currently HT is mainly used as bridging therapy for patients with ESLD awaiting OLT.

### 5.2. Stem Cell Transplantation

The limitations of hepatocyte transplantation require further studies on other alternatives. Stem cells could differentiate into hepatocyte-like cells (HLCs) in response to stimuli, which have become the most promising alternative strategy for cell transplantation of ESLD. HLCs can be differentiated from embryonic stem cells (ESCs) or from adult stem cells (AdSCs). Compared to ESCs, AdSCs enable autologous transplantation and generate less ethical concern [40]. AdSCs can be divided into liver stem cells (LSCs) and nonhepatic stem cells (Table 1).

#### 5.2.1. LSCs

LSCs are endogenous stem cells located in the liver, exhibiting self-renewing and bipotent properties. The liver originates from the foregut endoderm. During hepatic specification, the liver bud develops depending on hepatoblasts, which give rise to both hepatocytes and BECs. A population of hepatoblasts has been confirmed to have multipotent and highly proliferative features, which are termed as LSCs [41]. Another type of LSCs is LPCs, the bipotent stem cells located in the canals of Hering. LPCs have shown great prospect for liver regeneration. Lu et al. established a murine liver-damage model with the inducible deletion of E3 ubiquitin ligase Mdm2 in more than 98% of hepatocytes, which results in massive loss of hepatocytes. Then, LPCs isolated from wild-type mice were transplanted into these mice with hepatocyte injury due to genetic defect. In this study, the transplanted LPCs significantly contributed to restoration of liver parenchyma and functions, indicating LPCs as a potential alternative to hepatocyte or liver transplantation for liver diseases [42]. Numerous studies have confirmed that promoting LSCs to differentiate into hepatocytes might alleviate liver injury. However, this remains questionable due to the controversial question of whether the therapy will invoke tumorigenesis [41].

#### 5.2.2. Nonhepatic Stem Cells

The main types of nonhepatic stem cells for cell transplantation include induced pluripotent stem cells (iPSCs), mesenchymal stem/stromal cells (MSCs), hematopoietic stem cells, endothelial progenitor cells (EPCs), spermatogonial stem cells (SSCs), and others. 

ESCs are pluripotent stem cells, which have the potential to differentiate into the desired cell lineage, including hepatocytes. However, human ESCs are not yet readily used in clinics due to ethical concerns. iPSCs are generally derived from the reprogramming of mature somatic cells, which exhibit ESC features and advantages over ESCs for in vitro hepatocyte differentiation and maturation [4]. The breakthrough in generating iPSCs has not only overcomes the ethical issues and histological incompatibility associated with ESCs, but also enables correction of gene defects prior to cell transplantation [43]. iPSC-derived hepatocytes have great potential in regenerative medical therapies for ESLD. Furthermore, iPSC-derived hepatic organoids are beneficial in the field of regenerative medicine. However, cell therapies using iPSCs in the clinic is limited considering reprogramming efficiency and risk of tumorigenesis, due to reprogramming of somatic cells by gene transfer using viral vectors and their genetic instability [44].

MSCs are generally localized in umbilical cord blood, adipose tissue, cartilage and bone marrow. It is considered as one of the most effective multipotent cells, which is responsible for differentiating into hepatocytes, promoting hepatocytes’ proliferation, modulating immune and inflammatory responses, as well as regulating neovascularization [40,45]. Numerous studies have reported the therapeutic effects of MSCs’ transplantation on ESLD [46]. Recent studies mainly focus on modulation of MSCs to enhance the regeneration capacities as well as therapeutic effect. Enhanced PRL-1 expression in MSCs accelerates hepatic function via mitochondrial dynamics in a cirrhotic rat model [47]. A recent study revealed that VEGF165 overexpression enhances the multipotency of MSCs and promotes homing/colonization of MSCs in the liver tissues, which help to promote liver regeneration and ameliorated liver damage in rat models of acute liver failure [48]. In addition to cell transplantation, cell-free modalities relying on MSCs have been in investigation during recent decades. It is advocated that amnion-derived MSCs paracrine signals potentiate human liver organoid differentiation [49]. Suguru et al. advocated that in cirrhotic animals, small extracellular vesicles derived from IFN-γ preconditioned MSCs effectively induce anti-inflammatory macrophages and regulatory T cells, resulting in more efficient tissue repair. Potential tumorigenesis has been considered as a risk of MSCs due to their immunosuppressive and proangiogenic properties [50].

Hematopoietic stem cells originate in the embryonic liver, which are currently found in the bone marrow and umbilical cord blood. Apart from hematopoietic lineages, hematopoietic stem cells might differentiate into nonhematopoietic lineages, as indicated by the presence of hematopoietic stem cell-derived hepatocytes [40]. After partial hepatectomy in mice, hematopoietic stem cells are mobilized and recruited, which are responsible for limiting inflammation and boosting regeneration in a CD39-dependent manner [51]. However, transdifferentiation of hematopoietic stem cells in liver regeneration is challenging due to its extremely rare incidence [52]. Myerson et al. studied autopsy liver tissue of hematopoietic cell transplant recipients, suggesting that reconstitution of hepatocytes by hematopoietic stem cells result from infrequent fusion between incoming myelomonocytes and host hepatocytes, with subsequent proliferation [53]. 

EPCs are currently found in peripheral vessels and bone marrow. It has been demonstrated that recruitment of EPCs from the bone marrow by vascular endothelial growth factor-stromal cell-derived factor-1 (VEGF-sdf-1) signaling drives liver regeneration after partial hepatectomy in rats [54]. NG2-expressing cells are a population of periportal vascular progenitor cells (MLpvNG2(+) cells) isolated from healthy adult mouse liver. It is suggested that grafted MLpvNG2(+) cells would differentiate into hepatic lineages and restore liver function in mice with diethylnitrosamine-induced cirrhosis. In addition, grafted MLpvNG2(+) cells could mobilize endogenous stem cells to participate in liver regeneration [55].

SSCs are derived from testis. It is demonstrated that SSCs in vitro can transdifferentiate into cells with morphological, phenotypic and functional characteristics of mature hepatocytes via the activation of ERK1/2 and Smad2/3 signaling pathways and the inactivation of cyclin A, cyclin B and cyclin E [56]. Further study revealed that human SSCs can transdifferentiate into hepatocyte in CCl_4_-induced liver injury model of mice, providing a potential option for cell transplantation in patients with ESLD [57]. However, application of SSCs transplantation still requires further assessment in clinical studies.

A number of cell types have been under investigation for cell transplantation in the treatment of ESLD. It is suggested that human cord-lining epithelial stem cells (CLEC) can be differentiated into functional HLC. Raymond et al. addressed the safety of human CLEC transplantation in a porcine model of liver failure [58]. Biliary tree stem/progenitor cells (BTSCs) are derived from the peribiliary glands of the adult and fetal human biliary tree or from the crypts of the gallbladder, which can differentiate into hepatocytes, BECs and the islets of Langerhans cells. BTSCs have been shown to contribute to the renewal of extrahepatic biliary tree in mice upon damage [59]. Transplantation of bone marrow-derived endothelial progenitor cells together with hepatocyte stem cells from liver fibrosis rats significantly ameliorates liver fibrosis compared to transplantation of EPCs or hepatocyte stem cells alone [60]. These studies provide new prospects for treatment of ESLD.

### 5.3. Challenges and Future Prospects of Regenerative Medicine

Despite great development during recent decades, there are still multiple challenges regarding cell transplantation, and further studies are required. Long-term cell culture and passages might result in cytogenetic abnormalities. Over time, the transplanted cells may lose functional properties and even increase the risk of hepatic fibrosis as well as hepatocellular carcinoma [40]. To overcome these drawbacks, great effort has been made regarding preparation, modulation, delivery and encapsulation of the cell grafts. Advances in bioengineering and biomaterials have accelerated the development of regenerative medicine.

Reprogramming technology might help to resolve certain restrictions associated with cell transplantation, including malignancy potential and shortage of cell grafts. Yohan et al. reported that isolated human hepatocytes could be reprogrammed into human chemically derived hepatic progenitors (hCdHs) by two small molecules, A83-01 and CHIR99021, in the presence of EGF and HGF. hCdHs exhibited a significant potential of proliferation as well as differentiation into hepatocytes or BECs in vitro and in vivo, which would greatly facilitate regenerative medicine for treatment of ESLD [61]. Suhyun et al. reprogrammed mouse embryonic fibroblasts into induced hepatocyte-like cells (iHEPs) for the subsequent transplantation, which significantly attenuated liver fibrosis in CCl4-induced liver injury models [62]. Xie et al. advocated a two-step lineage reprogramming strategy to generate functionally competent human hepatocytes from fibroblasts [63].

Research on LPCs’ microenvironment has revealed the important roles of both biochemical and biomechanical signals in regulating LPCs functions. A recent study suggested an in vitro model of 3D LPCs spheroidal cultures with integrated polyethylene glycol hydrogel microparticles for the internal presentation of modular microenvironmental cues, which could be applied to the engineering of instructive microenvironments of transplanted LPCs. In this study, modification of the microparticles with TGF-β1 or heparin influences the behavior of LPCs toward BECs differentiation [64]. However, until now there has been limited data concerning engineering of 3D culture models of LPCs through the tunable presentation of microenvironmental stimuli.

The application of 3D bioprinting technology in tissue engineering enables the development of a biomimetic liver model that mimics the native liver module architecture. Ma et al. suggested a 3D hydrogel-based triculture model that embeds human induced pluripotent stem cells (hiPSCs)-derived LPCs with human umbilical vein endothelial cells (HUVECs) and adipose-derived stem cells in a microscale hexagonal architecture. The 3D triculture model shows significant superiorities in phenotype and functional properties [65].

The use of bioapplicable fabrication materials for cell transplantation has been proven as an attractive approach for the treatment of patients with ESLD. Yohan et al. constructed a patient-specific hepatic cell sheet from hCdHs on a multiscale fibrous scaffold by combining electrospinning and three-dimensional printing. Transplantation of the hepatic patch effectively repopulated the damaged parenchyma and restored liver function with an improved survival in mice. In this study, the functional properties of hCdHs were enhanced due to the histological and morphological similarity to liver tissue structure [66]. 

Organoid technology is a promising modality in regenerative medicine for the treatment of ESLD. It has been demonstrated that mass production of organoids is essential for treating various liver diseases since an organoid is more complex and functional than a single cell population [67]. 

## 6. Conclusions

ESLD is the end stage of various liver diseases and presents a worldwide health problem. As the only curative option for ESLD, OLT is associated with a series of limitations that restrict its clinical application. Concerning the prevalence of regeneration in liver injury, it is suggested that promoting liver regeneration should be beneficial to patients with ESLD. 

Liver regeneration is a complicated process that involves the cooperation of various cells and cytokines. Hepatocytes and BECs are homologous during embryonic development, which enables the potential of transdifferentiation between hepatocytes and BECs. The typical liver regeneration is achieved via the proliferation of pre-existing hepatocytes, whereas the alternative liver regeneration is achieved by the proliferation and subsequent differentiation of LPCs. During ESLD there is severe liver damage, and the proliferation of hepatocytes is greatly compromised; thus, the alternative liver regeneration is extremely important in ESLD. 

Advances in knowledge of liver regeneration and tissue engineering have greatly improved therapeutic strategies for ESLD. Regenerative medicine, especially cell transplantation, is a promising alternative to OLT in patients with ESLD. However, experimental models often fail to faithfully mimic human liver diseases. Furthermore, liver functions greatly rely on 3D assembling of hepatocytes with the supporting cells in a functional unit. The therapeutic effect of a simple transplantation with hepatocytes or stem cells was questioned due to the poor survival of cell grafts after transplantation. In conclusion, the gap from bench to clinic of regenerative medicine in patients with ESLD must be lessened. 

## Figures and Tables

**Figure 1 biomolecules-11-01907-f001:**
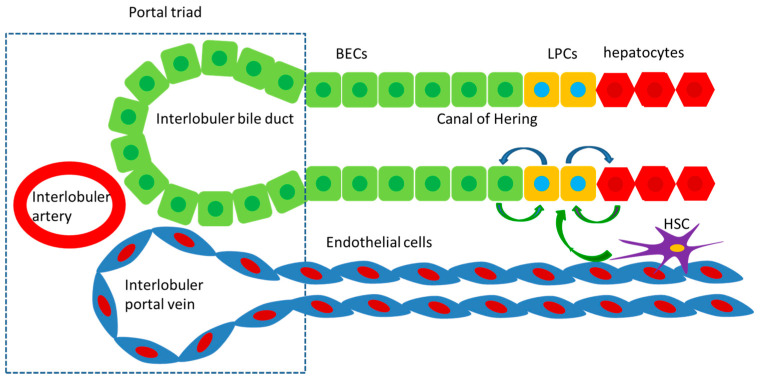
The alternative liver regeneration is driven by LPCs located in the canals of Hering, which might be derived from BECs, hepatocytes, and HSCs. LPCs can differentiate into BECs or hepatocytes to restore the liver mass and functions.

**Table 1 biomolecules-11-01907-t001:** Common types and location of AdSCs.

Cell Type	Location
LSCs	Liver
MSCs	Umbilical cord blood, adipose tissue, cartilage, bone marrow
Hematopoietic stem cells	Bone marrow, umbilical cord blood
EPCs	Peripheral vessels, bone marrow
SSCs	Testis

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
