# Peer review of "Liver Regeneration and Cell Transplantation for End-Stage Liver Disease"

_biomolecules, 2021, doi:10.3390/biom11121907_

Round 1

Reviewer 1 Report

  1. The authors chose a very interesting and challenging topi. The reviw successfully summarizes the current knowledge. However, some changes could help to clarify the message and add value to the manuscript.
  2. Topic 2: the authors should notify references
  3. Topic 3: briefly explain the "priming phase"
  4. Topic 3, Lines 91-95 and Lines 98-102: rephrase, the information is important but not clear
  5. Topic 3: it is important to clearly/explain in which condition we observe "typical" and in which "alternative liver regeneration. Indeed, the process is different after massive liver mass loss (partial hepatectomy) and adter massive liver injury.
  6. Topic 3+4+5: it is important to notify wherer the information/discoveries derive from experimental models or from human studies
  7. Topic 5: very interesting overview. However, it could add value in the review to specify the drawbacks/disadvantages/risks of each cell type cited in cell transplantation

Reviewer 2 Report

This is a review focusing on liver regeneration and cell transplantation. The article is interesting mainly for researchers. The authors discussed regeneration after liver resection. I think this theme would be more interesting for the readers. Please review the mechanism of liver regeneration after liver resection.

Round 2

Reviewer 2 Report

I think the authors has revised appropriately, and the article is ready for publication.